# When & How to Transfer with Transfer Learning

**Adrian Tormos**
Barcelona Supercomputing Center
adrian.tormos@bsc.es

**Dario Garcia-Gasulla**
Barcelona Supercomputing Center
dario.garcia@bsc.es

**Victor Gimenez-Abalos**
Barcelona Supercomputing Center
victor.gimenez@bsc.es

**Sergio Alvarez-Napagao**
Barcelona Supercomputing Center
sergio.alvarez@bsc.es

## Abstract

In deep learning, transfer learning (TL) has become the de facto approach when dealing with image related tasks. Visual features learnt for one task have been shown to be reusable for other tasks, improving performance significantly. By reusing deep representations, TL enables the use of deep models in domains with limited data availability, limited computational resources and/or limited access to human experts. Domains which include the vast majority of real-life applications. This paper conducts an experimental evaluation of TL, exploring its trade-offs with respect to performance, environmental footprint, human hours and computational requirements. Results highlight the cases were a cheap feature extraction approach is preferable, and the situations where an expensive fine-tuning effort may be worth the added cost. Finally, a set of guidelines on the use of TL are proposed.

## 1 Introduction

Training a deep learning (DL) model for image classification requires (1) powerful computational devices, (2) long execution times, (3) lots of energy, (4) large storage spaces, (5) data labelling efforts, and (6) time from human experts. Research has focused on mitigating these needs: Neural architecture search tackles 2, 3 and 6. Semi-supervised, weakly-supervised or self-supervised learning tackle 5. This paper focuses on transfer learning, which can mitigate all six requirements.

Transfer learning (TL) assumes representations learnt by a model can be reused for other tasks. The conditions and consequences of transferring parameters across tasks have been studied from the perspective of performance only [31, 18, 1, 25]. The general conclusion being that parametric reuse is better than starting from a random state, requiring less data for achieving higher generalisation.

In image classification tasks, the most common approaches of parametric reuse are feature extraction (FE) and fine-tuning (FT). In FE, the model's parameters are preserved statically, and the network's activations for the target task data are extracted and fed to another model. In FT, the model's parameters are further optimised for the new task, starting from their initial state. Both approaches have different pros and cons, and no clear guidelines exists to support a choice between both. This work fills that gap, by taking into account different perspectives, including the availability of pre-trained models on similar tasks, data volume, and the trade-offs between performance, carbon footprint, human cost and computational requirements.

Has it Trained Yet? Workshop at the Conference on Neural Information Processing Systems (NeurIPS 2022).

## 2    Related work

Transfer learning is thoroughly used to improve the development of DL models, as large-scale datasets have been released with the main purpose of producing pretrained models available for TL [28, 17]. Among the characteristics that positively influence the transferability, previous works have identified similarity between tasks [31], generality of the patterns learnt [31, 18], and size and variety of pretraining data [28, 17].

Beyond the pretraining model, target dataset size is also key. When little data is available, FT all the parameters of a large model easily leads to overfitting. One may freeze part of the parameters, or even discard them making the model shallower. Another way of preventing this is to use the pre-trained model to transform the data through its learnt patterns [7]. This FE is most commonly per-formed using features from a late layer (*e.g.,* the last or second to last layer of CNN models [25, 7]), and use those features to train linear models [7]. Since the depth of the layer may affect transferabil-ity, one may transfer the entire network representation while reducing its complexity without losing descriptive features (*e.g.,* through discretisation) like the Full-Network Embedding (FNE) [11].

## 3    Methodology

To assess the pros and cons of FT and FE we use publicly available pretrained models, applying them for the resolution of 10 different tasks. This work has been performed using the Tensorflow and Keras libraries, version 2.6.0. FT experiments have used an IBM Power9 8335-GTH processor and an NVIDIA V100 GPU; FE experiments have used an AMD EPYC 7742 processor. Footprint metrics have been estimated on a sample of experiments performed in an NVIDIA GeForce RTX 3090 GPU and an Intel Core i7-5820K CPU, tracking power with CodeCarbon[1]. Each experiment was performed once using a random seed.

For FT, we tune the model while freezing a variable number of initial layers, and randomising the last two layers. Training stopped when three consecutive epochs show a non-improving validation loss as early stopping policy. A minimum of 10 and a maximum of 25 epochs are computed. Batch size is 64. For FE we use the FNE, which extracts activations from a percentage of layers starting from the last before logits, using an average pooling operation on convolutional layers to extract one number per channel. Activations are feature-wise standardised and discretised to values in $\{-1, 0, 1\}$ [11]. Experiments are given a time limit of 24 hours. Data augmentation is used in both FT and FE, using 10 crops per sample (4 corners and central, with horizontal mirroring). During inference, crop predictions are aggregated using majority voting. Full code can be found in [2].

We use two VGG16 [27], one trained on *ImageNet 2012* [23] (*IN*, 1.2M train images) with Top-1/Top-5 accuracy of 71.3/90.1, the other trained on *Places 2* [33] (*P2*, 1.8M train images) with 55.2/85.0 accuracy. These are publicly available at [3] and [4], respectively. The tasks used for eval-uation are chosen to be representative of a variety of scenarios. Some are direct subsets ($\subset$) of a pretraining dataset, some intersect ($\cap$) with them, and some are entirely disjoint ($\varnothing$). Details can be found in Annex A.

To assess the trade-offs between FT and FE we use metrics from four categories. *(1) Performance*: Validation and test mean class accuracies ($V_{ACC}$, $T_{ACC}$), and overfitting ($V_{ACC} - T_{ACC}$). *(2) Foot-print*: power average in kW ($P_{AVG}$) in a sample of trainings, estimated greenhouse gas emissions in Kg of $CO_2$ ($E_{CO_2}$). *(3) Computational requirements*: execution time in hours ($T$), amount of experiments ($n_{EXP}$). *(4) Human cost*: Time analysing results and designing experimentation ($A$).

## 4    Hyperparameter search comparison

We perform model selection processes to compare the costs and benefits in the standard process of looking for a well-performing model. These are run for the 20 possible pretrained model and task pairs, and consider a few hyperparametric variables, chosen by their impact on performance. The

---

[1]https://codecarbon.io/

[2]https://github.com/HPAI-BSC/tl-tradeoff

[3]https://keras.io/api/applications

[4]https://github.com/CSAILVision/places365

Table 1: Performance ($V_{ACC}$ and $T_{ACC}$, average), footprint ($P_{AVG}$ average, $E_{CO_2}$ sum), computational requirements ($T$ sum, $n_{EXP}$ total) and human cost ($A$, sum) of the hyperparameter searches.

|  | $V_{ACC}$ | $T_{ACC}$ | $P_{AVG}$ | $E_{CO_2}$ | $T$ | $n_{EXP}$ | $A$ |
|---|---|---|---|---|---|---|---|
| FT | 77.46 | 73.86 | 276.1W | 201.54kg | 1,825.72h | 480 | 4-6h |
| FE | 74.65 | 72.73 | 124.1W | 3.84kg | 60.02h | 80 | 0-1h |

chosen hyperparameter values, the results for all metrics and the configuration with the best $V_{ACC}$ for each task are detailed in Annexes C.1 and C.2.

**Performance** In 8 out of 10 tasks, FT obtained the better performing model (mean of 2.84±8.66%). In overfitting ($V_{ACC} - T_{ACC}$), FT shows a slightly higher drop in test performance (mean drop of 3.6%, for FE's 1.92%).

**Footprint** The average power consumption in FT was 276.1W, 222% bigger than that of FE (124.1W). FT emitted 52.5 times more $CO_2$ than FE. In the context of the performance-footprint trade-off, notice this increase in footprint produces a mean improvement in $V_{ACC}$ of 2.81%. For $T_{ACC}$, only 1.13%.

**Computational requirements** The FT search lasted a total $1,825.72$ hours, $30.4$ times more than the FE search. In 9 out of 10 tasks, the FE search was faster than FT. Each FT search also required 6 times more experiments, influenced by a larger hyperparameter selection.

**Human cost** The time dedicated by experts on analysing results and designing experimentation ($A$), show how FE results are significantly easier to process thanks to its plain performance metrics. However, when performing FT, it is relevant to look at the tables beyond simple metrics, as part of the information regarding model convergence is lost when looking at a single performance metric. Thus, processing the results from FT entails the analysis of multiple training curves, which require in the order of 4 to 6 times more time. That being said, current deep learning frameworks are more oriented towards FT, which make FE slightly harder to implement.

The following section explores the role of training set size on performance, using the best hyperparameter configuration found before. However, such configuration may not remain competitive while using a different number of training samples. To verify this, we repeat the selection process for two subsets of the *Caltech 101* task, using only 5 and 10 samples per class – instead of the original 20 (sampling done to avoid the footprint reaching tons of $CO_2$). The results, detailed in Annex C.3, indicate FT is more brittle to changes in dataset size, requiring additional hyperparametric exploration when that happens. In this regard, some experiments from Section 5, may be using sub-optimal settings, and slightly underestimating the potential accuracy of FT approaches.

## 5   Few-shot learning

We study the effects of limited data availability, by subsetting the target datasets prior to training. For each target and number of instances per class ($IC$), we generate 5 random subsets (to mitigate statistical variance). We train FT and FE models and extract their final $T_{ACC}$. The distribution and mean relative difference in $T_{ACC}$ of the five runs for each $IC$ are shown in Figure 1. These results allow us to categorise datasets in two types: those where FT overtakes FE given enough samples per class, and those where FT and FE converge to the same accuracy. This distinction can be made at around 25 instances per class: if FT has not overcome FE with that amount or less, it will not happen regardless of data availability. Notice such threshold may depend on the architecture or source dataset employed [28, 17].

Figure 1 illustrates how, for those datasets that are disjoint from the pretraining dataset, FT and FE performance tends to converge. On the other hand, for those datasets which overlap (either intersect or subset) with the pretraining dataset, FT outperforms FE. We are unsure about the cause of this difference in behaviour. Another factor to consider here is time until convergence. While both methods scale linearly with $IC$ (see Annex D for details), the cost of FT grows 7 times faster than the cost of FE.

Let us remark that FT results may be slightly underestimated, as the hyperparameter search was not conducted for all $IC$ values. However, since the chosen hyperparameters are those that perform best

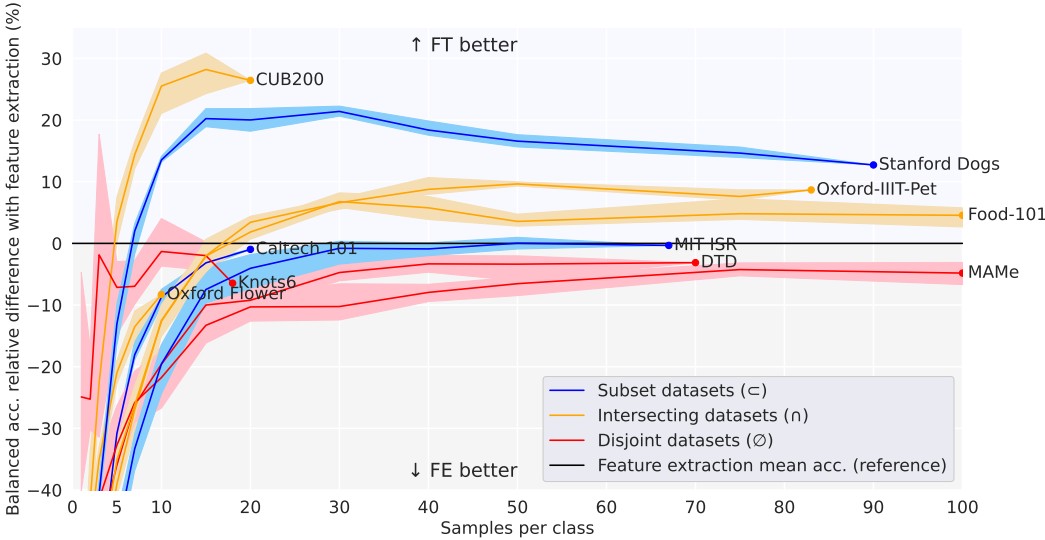

Figure 1: Relative difference in test accuracy ($100 \cdot \frac{\text{FT}-\text{FE}}{\text{FE}}$), *w.r.t.* the train split size. Shadowed regions show the minimum and maximum differences among the 5 random subsets. Black line represents point in which FE and FT have same performance.

on the highest $IC$ – the last point of each curve in Figure 1 – the final difference shown in Figure 1 remains reliable, while the intermediate slope may vary.

## 6    Discussion and future work

While FT is superior in *performance*, FE is better in terms of *footprint*, *computational requirements* and *human cost*. This is important in domains where TL is unfavourable (*i.e.,* one cannot find pretraining models close to the target task). Domains in which data availability is highly variable (*e.g.,* because of new data constantly being added) also favour FE, as FT requires for constant and expensive hyperparameter searches while FE does not.

The gain in performance provided by FT over FE is, on average, 2.8% on $V_{ACC}$, and 1.1% on $T_{ACC}$. Meanwhile, FT produces in the order of 7,000% more $CO_2$ than FE, and demands between 4 and 6 times more of human effort. It this context, researchers and practitioners of TL deciding between FT and FE should consider the relevance of limited performance gains for each application. While additional performance gains for FT over FE could be obtained by intensive data gathering, labelling and pretraining efforts, these would correspondingly increase cost FT, allowing this point to hold.

In few-shot learning scenarios (around 5 or less samples per class), the previous performance gain of FT vanishes, making FE the best performing model in most settings. Starting at samples per class of 5 and above, intersecting and subset ($\subset, \cap$) targets can have FT overtake FE by considerable amounts. For disjoint datasets FT may require more than 100 samples per class to outperform FE, while the latter remains as a competitive baseline. We believe these results will hold for architectures and pre-training datasets comparable to the ones used here (*i.e.,* CNNs and datasets in the order of a few millions of training samples). Beyond that, these experiments will need to be reassessed.

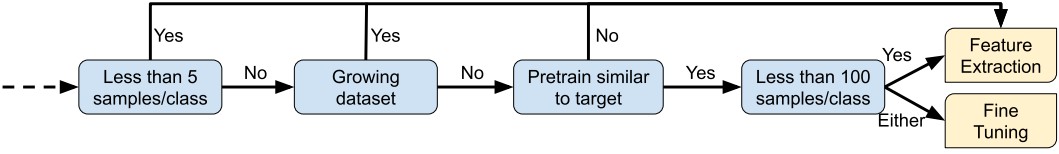

Figure 2: Flowchart for FE and FT recommendation, prioritising performance over other metrics. *Either* stands for both *Yes* and *No*, as FT is recommended regardless of the case.

As for specific hyperparametric findings, FT's best models are obtained by freezing 75% of layers. This is influenced by the quality and variety of internal features, and by the reduced number of trainable parameters, which prevents early overfitting. For FE, the percentage of extracted layers had a limited impact in performance (less than 2% on all cases). Thus, if marginal gains are irrelevant for the problem at hand, any of these settings is likely to provide competitive results. If margin gains are relevant, an exhaustive search can be conducted given the limited cost of the approach.

The scope of this work is limited to one architecture (VGG) and two pre-training datasets (*ImageNet* and *Places 2*), to limit the sources of variance. Extending the analysis would multiply the costs associated to this work (which are already high as seen in Table 4). At the same time, while related architectures or larger pre-training datasets may provide higher performance, this is likely to happen for both FT and FE, although the difference between performance gains may change the threshold values. As an additional contribution, Figure 2 includes a visual guide for the selection of model when prioritising performance metrics only.

## 7   Acknowledgement

We acknowledge the contributions of Ferran Parés and Armand Vilalta in the conceptualization and early iterations of this work. This work was partially supported by the AI4Media European project (Grant agreement 951911).

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

## A  Target tasks

The 10 tasks chosen for evaluation are separated in three categories. Some are direct subsets[5] ($\subset$) of a pretraining dataset, some intersect ($\cap$) with a pretraining dataset[6] but expand beyond it, and some are entirely disjoint ($\varnothing$) to the pretraining datasets.

Each dataset category represents one of the three scenarios one may face when doing TL in practice. Either you find a pretrained model which is a superset of the task at hand, one which at least intersects with it, or in the worst case scenario, one which is completely unrelated to it. The tasks, along with their state-of-the-art accuracy, are listed in Table 2.

The *Oulu Knots* dataset suffers from large inter-class imbalance. Of the seven original classes, the most underrepresented is 10% the size of the largest one. By removing the smallest of all classes we derive a second dataset, which allows us to further scale experimentation in Section 5. This modified dataset is referred as *Knots6*.

The *Caltech 101*, *MAMe* and *Oxford-IIIT-Pet* datasets have CC BY 4.0, CC BY-NC-ND 4.0 and CC BY-SA 4.0 licenses, respectively. No license has been found for other datasets.

Table 2: Basic features and state-of-the-art accuracies of target datasets. Those marked with (*) had no validation split, and their train split was divided in train and validation for this work. Those marked with (†) had a bigger validation split originally, which was reduced in favour of the train set. *Src* indicates the superset or intersecting source dataset.

|  |  | Images per class |  |  |  |  |  |
| --- | --- | --- | --- | --- | --- | --- | --- |
| Dataset | Classes | Train | Val | Test | Best Test Acc. (%) | Type | Src |
| Caltech 101* [8] | 101 | 20 | 10 | 1-50 | 94.4 [2] | $\subset$ | IN |
| CUB200* [30] | 200 | 20 | 10 | 11-30 | 92.3 [16] | $\cap$ | IN |
| DTD† [5] | 47 | 70 | 10 | 40 | 75.7±0.7 [14] | $\varnothing$ | - |
| Food-101* [3] | 101 | 190[7] | 10 | 250 | 90.4 [6] | $\cap$ | IN |
| MAMe [20] | 29 | 200[7] | 50 | 95-700 | 88.9 [20] | $\varnothing$ | - |
| MIT ISR* [22] | 67 | 67-73 | 10 | 17-23 | 90.3 [24] | $\subset$ | P2 |
| Oulu Knots [26] | 7 | 11-143 | 0 | 3-36 | 98.7 [10][8] | $\varnothing$ | - |
| Knots6* | 6 | 18-138 | 5 | 6-36 | - | $\varnothing$ | - |
| Oxford Flower [19] | 102 | 10 | 10 | 20-238 | 98.9 [13] | $\cap$ | IN |
| Oxford-IIIT-Pet* [21] | 37 | 83-90 | 10 | 88-100 | 95.4 [32] | $\cap$ | IN |
| Stanford Dogs* [15] | 120 | 90 | 10 | 50-152 | 92.9 [13] | $\subset$ | IN |

---

[5]Standford Dogs' labels are a direct subset of IN. Some Caltech 101 classes are not found in IN, but there are similar or visually analogous ones. For example, Caltech 101 has "metronome", "scissors" and "stapler". IN has "buckle", "knot", "nail", "screw", "hair slide clip", "combination lock", "padlock", "safety pin", among others.

[6]IN has less than 30 types of dishes for Food's 101, less than 60 birds for CUB's 200, and less than 10 flowers for Flowers's 102.

[7]Train splits of *Food-101* and *MAMe* were reduced from 740 to 190 samples and from 700 to 200, respectively.

[8]Reference performs data up-sampling for test inference, potentially biasing the accuracy. Alternative best accuracy is 74.1±6.9 [11].

## B  Carbon footprint computation

The $E_{CO_2}$ metric (estimated greenhouse gas emissions in Kg of $CO_2$) is approximated by using the ratio of $CO_2$ emissions from public electricity production provided by the European Environment Agency for the 27 European Union countries in 2020[9] (230.7 g/kWh).

## C  Hyperparameter search comparison

### C.1  Hyperparamter variables

The hyperparameter searches must be representative of real life model selection processes while having a reasonable scale. As such, we consider a few variables, chosen by their impact on performance. FT and FE hyperparameters are limited to 24 and 4 configurations, respectively. Note that there are more possible hyperparameter variables in FT, thus more hyperparameter variables than FE are explored. For FE, the classifier is fixed as a Linear SVM. For FT, the optimizer is fixed as SGD.

Table 3: List of tested hyperparameters on FT (up) and FE (down).

| Percentage (absolute num.) of frozen layers | Learning rate | Weight Decay | Momentum |
|---|---|---|---|
| 25% (4) / 50% (8) / 75% (12) | 0.01 / 0.001 | 0.001 / 0.0001 | 0.75 / 0.9 |

| Percentage (absolute num.) of extracted layers |
|---|
| 25% (3) / 50% (7) / 75% (11) / 100% (15) |

For FT, note that other minor hyperparameters have not been considered in an effort to limit the size of the search. Particularly, batch size has not been considered in order to avoid unnecessary overlap with the effect of learning rate [12].

In the case of FE, additional classifiers were initially considered such as RBF SVM, XGBoost [4] or KNN [9]. The latter two were discarded from the hyperparametric space because of non-competitive performance.

As for RBF SVM, the full model selection process were performed. It is the only variable we would remove from any future experimentation. The RBF experiments amounted for 923.63 hours, as opposed to 60.02 hours for LSVM. 32 out of 80 experiments did not even finish in the given 24 hours. In addition, they emitted 59.11kg of $CO_2$, as opposed to 3.84kg for LSVM. Coupled with the fact that LSVM has consistently (46 out of 60 experiments) better validation accuracy than RBF SVM, with the rest showing marginal improvements (1.04±0.72%), these results discourage the use of RBF SVM.

Full results for all hyperparameter searches can be found in Appendix B of [29].

Table 4: Performance ($V_{ACC}$, average), Footprint ($E_{CO_2}$ sum) and computational requirements ($T$ sum) of the FE hyperparameter searches, separated by classifier. If an experiment did not finish in the given 24h, that much time was added to the $T$ sum.

| | $V_{ACC}$ | $E_{CO_2}$ | $T$ |
|---|---|---|---|
| Linear SVM | 74.65 | 3.84kg | 60.02h |
| RBF SVM | 72.50 | 59.11kg | 923.63h |

### C.2  Performance results

In 8 out of 10 tasks, FT obtained the better performing model (mean of 2.84±8.66%). For those two tasks in which FE outperformed the difference was 0.45% and 5.69%. In overfitting ($V_{ACC} - T_{ACC}$), FT shows a slightly higher drop in test performance (mean drop of 3.6%, for FE's 1.92%).

The best pretraining dataset was IN, for 9 out of 10 targets in FE, and 8 out of 10 in FT. This aligns with similarity between pretraining model and target task. Regarding hyperparameters, the best FT setting consistently entails freezing the 75% first layers, even for those target tasks with the most training samples. In FE, the best setting ranges between extracting the activations of the first 50%, 75% and 100% of layers, although differences in performance are small.

---

[9]https://www.eea.europa.eu/ims/greenhouse-gas-emission-intensity-of-1

Table 5: Best hyperparametric configurations after model selection. *Src* stands for source model (ImageNet or Places2). *Layers* means extracted layers for FE, and frozen layers for FT. *LR* stands for learning rate. *WD* stands for weight decay. *Mom.* stands for momentum.

| Feature extraction | | | Fine-tuning | | | | | |
|---|---|---|---|---|---|---|---|---|
| Dataset | Src | Layers | Dataset | Src | Layers | LR | WD | Mom. |
| Caltech 101 | IN | 50% | Caltech 101 | IN | 75% | $10^{-3}$ | $10^{-4}$ | 0.75 |
| CUB200 | IN | 50% | CUB200 | IN | 75% | $10^{-2}$ | $10^{-4}$ | 0.75 |
| DTD | IN | 75% | DTD | IN | 75% | $10^{-3}$ | $10^{-4}$ | 0.9 |
| Food-101 | IN | 75% | Food-101 | IN | 75% | $10^{-2}$ | $10^{-4}$ | 0.75 |
| Knots6 | IN | 25% | Knots6 | IN | 75% | $10^{-2}$ | $10^{-3}$ | 0.75 |
| MAMe | IN | 100% | MAMe | P2 | 50% | $10^{-2}$ | $10^{-4}$ | 0.75 |
| MIT ISR | P2 | 75% | MIT ISR | P2 | 75% | $10^{-3}$ | $10^{-4}$ | 0.9 |
| O. Flower | IN | 100% | O. Flower | IN | 75% | $10^{-2}$ | $10^{-4}$ | 0.9 |
| O.-IIIT-Pet | IN | 75% | O.-IIIT-Pet | IN | 75% | $10^{-3}$ | $10^{-4}$ | 0.9 |
| Stanford Dogs | IN | 50% | Stanford Dogs | IN | 75% | $10^{-2}$ | $10^{-4}$ | 0.75 |

Table 6: Balanced accuracies in % for the FT and FE hyperparameter searches.

| | | Caltech 101 | CUB200 | DTD | Food-101 | Knots6 | MAMe | MIT ISR | O. Flower | O.-IIIT-Pet | S. Dogs | MEAN |
|---|---|---|---|---|---|---|---|---|---|---|---|---|
| FT | $V_{ACC}$ | 89.50 | 59.95 | 66.60 | 58.32 | 96.67 | 76.21 | 77.16 | 83.92 | 92.97 | 73.25 | 77.46 |
| | $T_{ACC}$ | 88.70 | 60.62 | 67.66 | 62.30 | 70.56 | 73.33 | 76.58 | 80.23 | 88.28 | 70.34 | 73.86 |
| FE | $V_{ACC}$ | 89.01 | 48.45 | 65.74 | 52.28 | 93.33 | 74.48 | 77.61 | 89.61 | 91.35 | 64.25 | 74.65 |
| | $T_{ACC}$ | 89.57 | 47.95 | 69.84 | 58.73 | 75.43 | 77.85 | 76.83 | 87.48 | 81.24 | 62.40 | 72.73 |

## C.3   Variance of scale in model selection

In Section 5 we explore the role of training set size using the best hyperparameter configuration found in Table 5. However, we cannot assure such configuration remains the best while using a different number of training samples. To verify to which extent the best configuration holds, without recomputing the full hyperparametric search between 6 and 15 times (with footprint reaching tons of $CO_2$), we repeat the selection process for two subsets of the *Caltech 101* task, using only 5 and 10 samples per class – instead of the original 20.

Table 7: Best hyperparametric configurations after model selection with train split subsets of *Caltech 101*. *IC* stands for instances per class. The best pretraining source was always *IN*. *Drop* expresses the difference in $V_{ACC}$ of the selected configuration in the original search (20 samples/class) when trained with smaller train subsets *w.r.t.* the best hyperparameter configuration selected for that subset. *Src* stands for source model (*ImageNet* or *Places 2*). *Lay.* stands for extracted layers for FE, and frozen layers for FT. *LR*, *WD* and *Mom.* stand for learning rate, weight decay and momentum.

| Feature extraction | | | | Fine-tuning | | | | | | |
|---|---|---|---|---|---|---|---|---|---|---|
| *IC* | Lay. | $V_{ACC}$ | Drop | *IC* | Lay. | LR | WD | Mom. | $V_{ACC}$ | Drop |
| 5 | 75% | 81.12 | 0.07 | 5 | 75% | $10^{-2}$ | $10^{-3}$ | 0.9 | 76.24 | 17.92 |
| 10 | 50% | 85.84 | 0.00 | 10 | 75% | $10^{-2}$ | $10^{-4}$ | 0.9 | 83.96 | 6.34 |
| 20 | 50% | 89.01 | – | 20 | 75% | $10^{-3}$ | $10^{-4}$ | 0.75 | 89.50 | – |

For FE, all hyperparameters had the same behaviour in training with both 5 and 10 samples per class, than in the original search. With only one exception: extracting 50% of layers instead of 75% provided a marginal gain of 0.07% in accuracy for the 5 samples case.

For FT, some hyperparameters are invariant to the change in training set size, such as the best pretraining source and amount of frozen layers (where 50% and 75% remain better than 25%). The optimiser hyperparameters, however, depend on the amount of data available. When less than 20 samples are available, the best *LR* changes (from $10^{-3}$) to the largest one ($10^{-2}$), and when using

5 samples per class, the previously best performing *WD* ($10^{-4}$) has a drop in validation accuracy around 20% (*w.r.t.* $10^{-3}$); possibly due to the maximum epochs imposed for model selection experiments. These results indicate FT is more brittle to changes in dataset size, requiring additional hyperparametric exploration when that happens. In this regard, the analysis of Section 5, may be using a sub-optimal setting, and slightly underestimating the potential accuracy of FT approaches.

## D    Few-shot learning computational resources

Both FE and FT generally scale linearly with the amount of instances per class. However, the latter has a larger overhead and its cost scales up to 7 times faster than the former. Figure 3 shows how both methods scale for each of the targets.

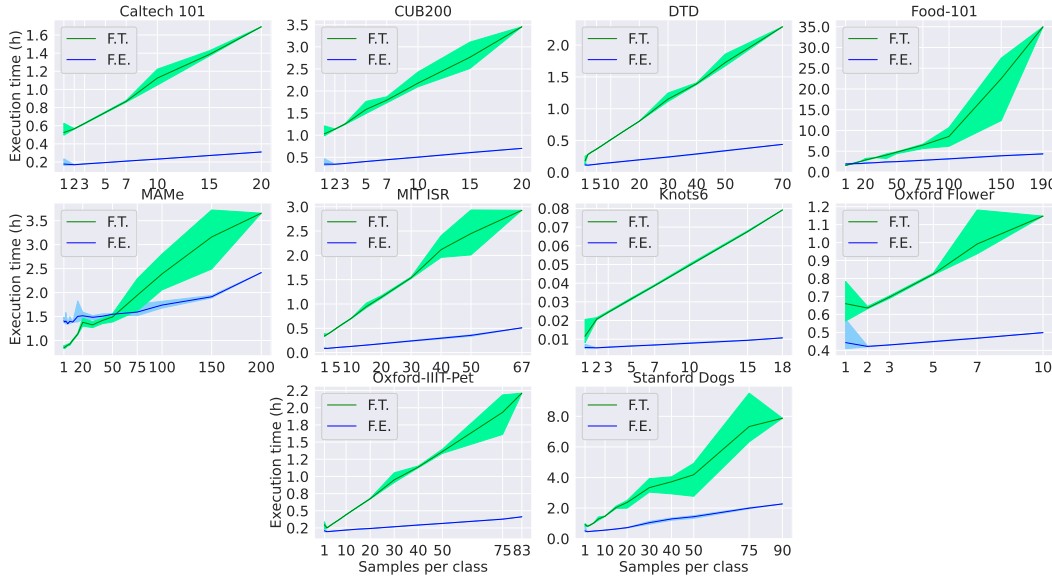

Figure 3: FT and FE time until convergence (h) in 4 datasets, with respect to instances per class. Mean (line) and minimum/maximum (area) times are given, computed across random subsets. Time from the first image loading to the final validation and test accuracy reporting.

