# OpenReview forum: "When & How to Transfer with Transfer Learning"
_NeurIPS.cc/2022/Workshop/HITY — HITY Workshop NeurIPS 2022_

### Official Review · Reviewer_qUyB · 2022-10-11
**Practically useful paper**

**Rating:** 1
**Confidence:** 4

**Review:**

This paper compares the transfer learning approaches "feature extraction" and "fine-tuning". Several interesting metrics are considered, including: performance, power consumption, $CO_2$ emissions, execution time,  and human costs.

It is shown that fine-tuning is better for performance, but feature extraction is better in $CO_2$ emissions, execution time, and human costs.


The paper is clearly written and understandable throughout. The topic fits well to this workshop. The methodology is described in detail.

Drawbacks are:
- Results are not surprising.
- It is not clear how the human costs are determined. Did you average over several humans?
- Detailed results for each pre-trained model are not provided. Only the average over all models is considered. It would be interesting to know if there are any outliers and what the distribution looks like.
- It is unclear if the results for each task configuration were averaged over multiple seeds.

---

### Official Review · Reviewer_11Tn · 2022-10-12

**Rating:** 1
**Confidence:** 3

**Review:**

**Summary:** The paper empirically compares two approaches for transfer
learning: FE (feature extraction) and FT (fine-tuning). This comparison includes
multiple tasks and different metrics (e.g. performance and CO2 footprint). The
authors also investigate the impact of the amount of available data.

**Strengths, Weaknesses & Questions:**
- Overall, the paper is well-written. The methodology is clearly explained and
the authors also discuss limitations of their work. The experimental details are
described in the appendix.
- Due to my very limited experience with FE and FT, I would have appreciated a
more detailed explanation of the two methods.
- Line 53: For the FT approach, SGD with a fixed batch size of $64$ is used.
This parameter can have a crucial impact on the resulting performance, but this
is neither discussed nor investigated in the paper.
- Line 71-75: I am skeptical whether this comparison is fair since you are
using 24 hyperparameter configurations for FT and only 4 configurations for the
FE approach. It is not clear to me to which extent this imbalance biases your
results in favor of FT.
- Line 86-90: The definition of *human cost* is not clear to me: What exactly do
you mean by *analysis of multiple training curves*? Can't the extraction of the
relevant information be fully automated?
- Line 103-107: I don't fully understand this section. For example, the Oxford
Flower data set does not match the two categories you mention. I also don't
understand the *distinction at around $25$ instances per class*. Left and right
of this threshold, there are cases where FT outperforms FE.
- Line 110: Do you have an explanation for why FT outperforms FE on data sets
with overlap?
- Line 129: This statement is a bit too strong in my opinion. FT still
outperforms FE in 4 cases (by a lot), has almost identical performance in 2
cases (Caltech 101 and MIT ISR), and is slightly worse than FE in 4 cases.
- Line 130-133: I'm confused by these statements. For under $7$ training samples
per class, almost all data sets show negative y-values indicating that FE outperforms FT. This is in contradiction with your statement.
- Line 133-135, 144-146: I find these statements questionable and quite
speculative.

**Minor:**
- Typos: Line 10: *a* $\rightarrow$ *an*, Line 100: *buy* $\rightarrow$ *by*
- Line 58: You could have included a link to an anonymous GitHub repo with your
code.
- Figure 1: A text within the plot like *FT better* for positive y-values and
*FE better* for negative values would allow faster orientation.

---

### Official Review · Reviewer_SQ7x · 2022-10-17

**Rating:** 1
**Confidence:** 4

**Review:**

This paper studies transfer learning, and in particular studies how full fine-tuning compares against feature extraction in different settings, resulting in insightful advice on which approach to use for what kinds of problems. The paper is well written and easy to follow, and presents results that will be useful to the community.

---

### Decision · Program_Chairs · 2022-10-20

Accept